# Bromelain Improves Hypothalamic Control of Energy Homeostasis in High-Fat Diet-Induced Obese Rats

**DOI:** 10.3390/cimb47080607

**Published:** 2025-08-01

**Authors:** Raviye Ozen Koca, Mustafa Berk Basaran, Hatice Solak, Zulfikare Isik Solak Gormus

**Affiliations:** 1Department of Physiology, Faculty of Medicine, Necmettin Erbakan University, 42090 Konya, Turkey; mbebasaran@gmail.com; 2Department of Physiology, Faculty of Medicine, Kütahya Health Sciences University, 43100 Kütahya, Turkey; hhaticesolak@gmail.com

**Keywords:** obesity, bromelain, high fat diet, hypothalamus, energy homeostasis

## Abstract

Obesity remains a major global health challenge with limited therapeutic options. Bromelain, a proteolytic enzyme complex derived from pineapple, has been recognized for its natural anti-inflammatory, anti-edematous, and appetite-suppressing properties. This study aimed to investigate the effects of bromelain on hypothalamic neuropeptides and metabolic markers in a high-fat diet (HFD)-induced obesity model in rats. Thirty-six male Wistar albino rats were randomly divided into four groups: standard diet (SD), standard diet with bromelain (SDBro), high-fat diet (HFD), and high-fat diet with bromelain (HFDBro). Obesity was induced by a 3-month HFD regimen, followed by bromelain supplementation (200 mg/kg/day, orally) for one month. Hypothalamic tissues were analyzed via ELISA for neuropeptide Y (NPY), pro-opiomelanocortin (POMC), glucose transporter 2 (GLUT2), fibroblast growth factor 2 (FGF2), and insulin-like growth factor 1 receptor (IGF1R). While NPY levels showed no significant changes, POMC increased in the HFD and was normalized with bromelain. GLUT2 was downregulated in the HFD and significantly restored by bromelain. FGF2 levels remained unchanged. IGF1R was upregulated in the HFD but reduced by bromelain, with an unexpected increase in SDBro. Overall, bromelain partially reversed HFD-induced disruptions in hypothalamic energy-regulating pathways, particularly affecting GLUT2 and POMC. These findings highlight bromelain’s potential role in central metabolic regulation under dietary stress.

## 1. Introduction

Obesity is a chronic metabolic disorder characterized by the excessive accumulation of adipose tissue which adversely impacts health, and it is typically defined by a body mass index (BMI) of 30 kg/m^2^ or greater [1]. The global burden of obesity is escalating, owing to suboptimal treatment outcomes and its association with comorbidities such as anxiety, cardiovascular diseases, and metabolic syndromes. The hypothalamus and hippocampus are neurogenic brain regions implicated in the regulation of feeding behaviors and emotional states. Although the neurogenic potential of the hippocampus in response to dietary and lifestyle interventions has been well characterized, similar mechanisms in the hypothalamus remain poorly understood. Obesity induces a neurogenic shift toward orexigenic NPY-expressing neurons, disrupting energy balance [2]. Moreover, caloric restriction can reprogram orexigenic pathways, often resulting in hyperphagia upon diet cessation [3].

Recent evidence suggests that the adult hypothalamus constitutes a neurogenic niche, where dietary factors, particularly a HFD, provoke inflammation and disrupt neurogenesis, ultimately altering body weight regulation [4]. These findings highlight the potential for pharmacological interventions to modulate hypothalamic activity and restore metabolic homeostasis. Bromelain is a proteolytic enzyme complex derived primarily from the stem of Ananas comosus, exhibiting anti-inflammatory, anti-diabetic, anti-neoplastic, and anti-rheumatic properties [5]. Its anti-obesogenic actions include the downregulation of adipogenic genes and induction of apoptosis in mature adipocytes [6], as well as the improvement of lipid metabolism [7].

Bromelain exists in two primary forms: fruit bromelain (FBM) and stem bromelain (SBM), with SBM being more potent and economically viable due to its higher concentration and ease of extraction [8,9]. Animal studies support its safety profile, with a reported LD_50_ exceeding 10 g/kg of bromelain in rodents [10].

Despite its systemic effects, the impact of bromelain on hypothalamic neuropeptides involved in energy regulation remains to be elucidated. To our knowledge, no studies have evaluated bromelain’s influence on neurochemical markers within the hypothalamus in an experimental model of diet-induced obesity. We hypothesized that bromelain may modulate hypothalamic peptide expression and metabolic signaling pathways. Accordingly, this study aimed to evaluate the central metabolic effects of bromelain in a rat model with a HFD-induced obesity by quantifying the expression of key hypothalamic markers.

## 2. Materials and Methods

This study was approved by the Local Ethics Committee for Animal Research at the Experimental Medical Research and Application Center, NEU (Approval No: 2024-108, Date: 28 November 2024). All procedures were performed in accordance with institutional and national ethical guidelines for the care and use of laboratory animals. Statistical analyses were performed using Microsoft Excel for Office 365 (Microsoft Corporation, Redmond, WA, USA) and GraphPad Prism version 5.0 (GraphPad Software Inc., San Diego, CA, USA).

Thirty-six male Wistar albino rats, aged six weeks and weighing between 150 and 200 g, were used in the study. The animals were housed under standard laboratory conditions (12 h light/dark cycle, ambient temperature 21–23 °C, and 45–55% humidity) with ad libitum access to food and water.

### 2.1. Establishment of the Obesity Model and Bromelain Administration

Rats were randomly divided into four experimental groups (*n* = 9 per group): standard diet (SD), standard diet with bromelain (SDBro), high-fat diet (HFD), and high-fat diet with bromelain (HFDBro). Obesity was induced over a 3-month period in the HFD and HFDBro groups using a high-fat feed (Arden Research & Experiment Company, Ankara, Turkey). According to the manufacturer’s nutritional analysis, this diet consisted of 24% protein, 30% carbohydrates, and 35% fat (primarily vegetable oil), providing 5.2 kcal/g total energy, of which approximately 45% was derived from fat.

The SD and SDBro groups received standard rat chow. During the process of establishing the obesity model, the weights of the rats in each group were regularly monitored. The initial weights of the rats were recorded, and measurements were taken at approximately 10-day intervals (Figure 1).

After the obesity model was established, rats in the SDBro and HFDBro groups were administered bromelain for 30 days via oral gavage at a dose of 200 mg/kg/day [11]. The bromelain preparation, with a digestive activity of 2400 GDU/g (Meteoric Biopharmaceutical, lot number: BM2403332), was dissolved daily in physiological saline for administration. Each rat received 1 mL/100 g body weight (200 mg/kg) by oral gavage using a 16-gauge flexible feeding needle. Dosing was performed each morning (09:00–10:00 h) in a post-absorptive state. A single researcher gently restrained each rat, inserted the gavage needle, and administered the full dose while visually confirming swallowing. Any instance of regurgitation prompted an immediate repeat dose, and all events were logged to ensure dose delivery. Rats in the SD and HFD groups similarly received 1 mL/100 g body weight of physiological saline by gavage on the same schedule to control for handling and vehicle effects.

### 2.2. Body Weight, Fasting Glucose, and Tissue Collection

Body weight was recorded at baseline (Day 0) and every 10 days during the 90-day diet period. Rats on the HFD achieving ≥20% weight gain relative to the SD were classified as obese.

On Day 100, following a 12 h fast (18:00–06:00 h), a drop of tail vein blood was collected from each rat. Fasting blood glucose was measured in duplicate using a glucometer previously validated for rodent studies.

Immediately after blood sampling, rats were weighed to assess end-of-treatment body weights, then anesthetized with ketamine/xylazine (80/10 mg/kg, i.p.) to minimize suffering. Animals were euthanized by decapitation, and brains were rapidly removed. The hypothalamus was dissected on ice [12], snap-frozen in liquid nitrogen, and stored at –80 °C until biochemical analysis.

### 2.3. Biochemical Analyses

For tissue preparation, hypothalamic samples were homogenized in ice-cold phos-phate-buffered saline (PBS, pH 7.4) containing a protease inhibitor cocktail to prevent protein degradation by ELISA kits (Bioassay Technology Laboratory (BT LAB), Shanghai, China) in accordance with the manufacturer’s instructions. Homogenization was performed using a mechanical homogenizer under controlled conditions to ensure uniform sample consistency. The homogenates were centrifuged at 12,000× *g* for 15 min at 4 °C, and the supernatants were collected for analysis [13].

Quantitative measurement of the target protein’s IGF1R, (Bioassay Technology Laboratory [BT LAB], catalog no: E1719Ra); GLUT2, (Bioassay Technology Laboratory [BT LAB], catalog no: E1058Ra); POMC, (Bioassay Technology Laboratory [BT LAB], catalog no: E1130Ra); NPY, (Bioassay Technology Laboratory [BT LAB], catalog no: E0540Ra); and FGF2, (Bioassay Technology Laboratory [BT LAB], catalog no:E0852Ra) was performed using commercially available enzyme-linked immunosorbent assay (ELISA) kits, following the manufacturers’ protocols. Briefly, standards and samples were added to pre-coated microplate wells, incubated with specific antibodies, and subjected to washing steps to remove unbound substances. After the addition of the substrate solution, absorbance was measured at the specified wavelength using a microplate reader. Protein concentrations were calculated based on standard curves generated with the known concentrations of each analyzed. All samples were assayed in duplicate to ensure reproducibility.

All biochemical assays and data analyses (including ELISA measurements and glucose readings) were performed by investigators who were blinded to the animals’ group assignments.

### 2.4. Statistical Analysis

Normality of data distribution was assessed using the Shapiro–Wilk test. Group comparisons were conducted using a one-way analysis of variance (ANOVA) followed by Tukey’s post hoc test. Repeated measures of the one-way ANOVA were used to analyze within-group changes in body weight over time. Statistical analyses were performed using Microsoft Excel for Office 365 (Microsoft Corporation, Redmond, WA, USA) and GraphPad Prism version 5.0 (GraphPad Software Inc., San Diego, CA, USA).

## 3. Results

### 3.1. Body Weight Changes

Throughout the study, the body weight measurements of animals from different groups were evaluated. Body weights were recorded at nine time points throughout the experimental period. In the SD group, the first, second, and eighth measurements were found to be significantly different compared to the previous measurements (*p* < 0.05). In the HFD group, the first, second, fourth, and ninth measurements were found to be significant (*p* < 0.05). In the SDBro group, the first, second, third, fourth, and ninth measurements were observed to be statistically significant. In the HFDBro group, the first and second measurements were significantly different from the previous ones. In the HFD group, the values from the fourth to the ninth measurement were statistically significant when compared to the SD group. In the SDBro group, the sixth, seventh, eighth, and ninth measurements were significantly different compared to the HFD group. In the HFDBro group, the values from the first to the ninth measurement were significantly different compared to the SD group; the sixth and eighth measurements were significant compared to the HFD group; and the sixth, seventh, and eighth measurements were statistically significant compared to the SDBro group (*p* < 0.05). It was observed that the HFD group had the highest body weight, while the HFDBro group had a lower body weight compared to the HFD group (Table 1).

### 3.2. Fasting Blood Glucose Levels

Fasting blood glucose levels were significantly elevated in both the HFD and HFDBro groups relative to the SD and SDBro groups (*p* < 0.05), suggesting that a high-fat diet impaired glucose regulation, while bromelain supplementation alone did not normalize glucose levels in the HFDBro group (Figure 2).

### 3.3. Biochemical Parameters

The expression levels of IGF1R, GLUT2, POMC, NPY, and FGF2 were examined in hypothalamic tissues. No statistically significant differences were detected in hypothalamic NPY and FGF2 levels across the groups. However, POMC levels were found to be statistically significant in the SDBro group compared to the SD group and in the HFDBro group compared to both the HFD and SDBro groups (*p* < 0.05). GLUT2 levels were statistically significant in the HFD group compared to the SD group, in the SDBro group compared to both the HFD and SD groups, and in the HFDBro group compared to the HFD, SD, and SDBro groups (*p* < 0.05). IGF1R expression was significantly elevated in the HFD group compared to the SD, while bromelain supplementation in the HFDBro group led to a significant reduction in IGF1R levels compared to the SDBro group (*p* < 0.05). These findings collectively indicate that bromelain exerts differential regulatory effects on hypothalamic energy sensors, particularly GLUT2 and IGF1R, under conditions of diet-induced obesity (Figure 3).

## 4. Discussion

To develop effective treatments for obesity, new research avenues must be explored. In a previous study conducted by the project team, the effects of bromelain on the cardiovascular system in an experimental obesity model were investigated.

The hypothalamus is the primary regulator of energy balance in the body. First-order hypothalamic neurons located in the arcuate nucleus detect systemic signals that indicate the body’s energy stores. These neurons in the arcuate nucleus communicate via distinct projections primarily with second-order neurons located in the paraventricular nucleus and lateral hypothalamus. The signals are then transmitted to third- and fourth-order neurons, which activate complex responses aimed at maintaining overall energy homeostasis [14]. Metabolic sensors play a key role in regulating food intake, nutrient utilization, and feeding behavior [15]. Dietary glucose affects metabolic activity in hypothalamic regions that regulate energy homeostasis. The disruption of this mechanism may contribute to the development of obesity [16].

NPY is a highly conserved neuropeptide that plays a role in regulating energy homeostasis and exerts orexigenic effects in hypothalamic nuclei. NPY signals through a family of high-affinity receptors that mediate its widespread effects throughout the hypothalamic nuclei. These effects are also regulated by numerous peripheral and central energy balance signals. NPY directly stimulates food intake and reduces energy expenditure through Y1 and Y5 receptors [17]. In obese rats, NPY mRNA levels have been found to be significantly elevated in the medial hypothalamus [18].

In the literature, the effects of a high-fat diet on NPY expression are inconsistent. In our study, although no statistically significant differences were observed between the groups in terms of NPY levels, a trend toward an increased expression was noted in the HFD group compared to the SD group. It has been reported that hypothalamic NPY expression may increase in mice fed a high-fat diet; however, this increase may vary depending on experimental conditions and the duration of the diet [19]. Similarly, it has been suggested that a HFD may initially enhance NPY expression, but this effect may diminish with prolonged exposure to a high-fat diet [20]. The lack of a significant difference in our study may reflect this variability. Furthermore, factors such as the type of fat used (plant-based or animal-based), the duration of dietary intervention, and individual metabolic responses are thought to influence these results.

Hypothalamic POMC neurons integrate ambient glucose levels and other metabolic signals to maintain energy homeostasis [21]. POMC neurons signal satiety by activating melanocortin receptors [22], and their dysfunction leads to obesity and insulin resistance [23]. Chronic consumption of a HFD may impair the ability of the hormone leptin to activate POMC neurons, thereby weakening their anorexigenic responses [24]. The increase in POMC expression observed in the HFD group may be associated with disruptions in leptin signaling. The elevated POMC expression in the SDBro group compared to the SD group may be related to the anti-inflammatory properties of bromelain. By reducing hypothalamic inflammation, bromelain may support the function of POMC neurons. In the HFDBro group, the observed decrease in POMC expression suggests that bromelain may help restore normal POMC neuron function by alleviating inflammation and improving leptin signaling. In our study, the increase in POMC expression in the high-fat diet group indicates that such a diet disrupts hypothalamic energy homeostasis and affects satiety signaling. The reduction in POMC expression following bromelain supplementation, particularly in the high-fat diet group, may be linked to the enzyme’s anti-inflammatory effects and its role in regulating energy metabolism.

Central GLUT2 is essential for hypothalamic glucose sensing and energy balance, much like its role in pancreatic β-cells [16,18]. Chronic HFD feeding reduces the GLUT2 expression in key hypothalamic regions, demonstrated in obese Zucker rats, which impairs neuronal glucose uptake and sensing [18]. This “glucose resistance” disrupts normal AMPK signaling upon refeeding, contributing to defective satiety cues and overall energy homeostasis disorders in diet-induced obesity [16].

Hypothalamic cells co-express GLUT2 and glucokinase, highlighting their coordinated role in neuronal glucose sensing [15]. Suppression of GLUT2 in glial cells impairs satiety signaling and promotes increased food intake [25]. This finding suggests that bromelain may exert a regulatory effect on energy metabolism. It has been shown that pineapple juice containing bromelain reduces body weight gain, decreases hepatic lipid accumulation, and improves metabolic health in obese rats [26]. These effects are thought to be related to bromelain’s ability to enhance GLUT2 expression and thereby support central glucose sensing. In our study, the decrease in GLUT2 levels in the HFD group suggests weakened glucose signaling and sensing, leading to a disrupted energy balance. Conversely, the increase in GLUT2 levels following bromelain supplementation in both the SD and HFD groups indicates the potential of this proteolytic enzyme to improve central glucose sensing. This increase may contribute to the restoration of energy homeostasis.

In the adult hypothalamus, the neuronal progenitor role is attributed to radial glia-like cells called tanycytes, which line the wall of the third ventricle. Under nutritional cues, including hypercaloric diets, tanycytes proliferate and differentiate into mature neurons that regulate body weight, suggesting that hypothalamic neurogenesis is an adaptive mechanism in response to metabolic changes. FGF2 enhances β-tanycyte proliferation by triggering purinergic signaling and highlights certain molecular mechanisms involved in tanycyte cell division responses [15,27]. It is believed that tanycytes play a crucial role in integrating central and peripheral signals to regulate energy intake and expenditure in the hypothalamus. The expression of the FGF2 receptor FGFR1 in the ventral tanycyte region has been emphasized for its importance in controlling body weight and food intake [28]. FGF2 stimulates the proliferation of specialized radial astrocytes known as tanycytes along the hypothalamic ventricle and influences metabolism. It has been reported that a HFD and loss of the FGFR1 gene lead to an altered β and α tanycyte morphology and a reduced number of stem cells in the hypothalamus [29]. According to our findings, the lack of a significant effect of bromelain on hypothalamic FGF2 levels may reflect that bromelain does not directly target the FGF2 signaling pathway, that distinct tanycyte subpopulations exhibit differential sensitivity, or that experimental variables limited our ability to detect subtle changes. These inconclusive results indicate that dedicated studies focusing on tanycyte biology and alternative neurogenic markers are needed to clarify bromelain’s impact on hypothalamic neurogenesis.

IGF-1 is a polypeptide hormone with a high structural similarity to insulin and binds with high affinity to the IGF-1R, activating both mitogen-activated protein (MAP) kinase and phosphoinositide 3-kinase (PI3K) signaling pathways in target tissues [30]. Growth hormone (GH) and IGF-1 have definitive roles in regulating somatic development and play both direct and indirect roles in metabolic homeostasis and body growth [31]. Early postnatal undernutrition in rodents causes lasting growth retardation and reduced circulating IGF-1, alongside impaired GH/IGF-1 feedback sensitivity [32,33]. In obesity, diminished IGF-1 signaling is linked to dyslipidemia, hypertension, insulin resistance, type 2 diabetes, and cardiovascular disease [34]. IGF-1 directly drives preadipocyte differentiation, clonal expansion, and lipid droplet formation [35], while adipose IGF-1R exerts negative feedback on IGF-1 gene expression to adjust systemic levels and regulate somatic growth [36]. Furthermore, restoration of GH/IGF-1 signaling modulates hepatocyte lipid metabolism, enhances stress resilience, and regulates non-parenchymal cell-mediated inflammation and fibrosis [37].

Molecular interactions, metabolism, and biological activity findings have demonstrated that bromelain is a potential alpha-glucosidase inhibitor. Bromelain has been suggested as a useful insulin-independent therapeutic molecule for the control and management of type 2 diabetes mellitus [38]. The anti-diabetic property of bromelain arises from improvements in glucose metabolism and reductions in insulin resistance [5]. IGF1R levels were observed to be higher in the HFD group compared to the SD group. This finding suggests that HFD may increase IGF1R levels, which could adversely affect metabolic homeostasis. In the HFDBro group, however, a decrease was observed compared to the HFD group. This indicates that bromelain may suppress the HFD-induced increase in IGF1R, thereby contributing to metabolic regulation. Conversely, IGF1R levels increased in the SDBro group compared to the SD group. IGF1R plays an important role in metabolic homeostasis, and its effects may vary depending on factors such as diet-induced obesity and aging [39]. Additionally, IGF1R expression in the brain has been shown to vary depending on the caloric content and macronutrient composition of the diet, which may influence metabolic processes [40]. In this context, the effects of bromelain on IGF1R expression may depend on the type of diet and could play a significant role in metabolic regulatory processes.

### Limitations

This study was conducted using only male rats, and potential sex differences were not evaluated. Additionally, behavioral tests and advanced protein analyses (e.g., Western blot, immunohistochemistry) could not be performed due to budget constraints. This limitation restricted the functional and structural validation of the molecular findings obtained. Future studies are warranted to investigate the mechanisms of bromelain action in greater detail, including comparisons of different doses, durations, and sexes.

## 5. Conclusions

The present study aimed to evaluate the ability of bromelain to regulate feeding and affect energy homeostasis through hypothalamic mechanisms in obesity caused by chronic high-fat intake. It has been shown that a high-fat diet disrupts hypothalamic glucose sensing and energy balance, particularly through metabolic markers such as GLUT2, NPY, and POMC. Bromelain supplementation was observed to partially restore these disruptions and contribute to the re-establishment of neurochemical balance. These preclinical findings also raise the possibility that bromelain could be incorporated into functional foods or used as an adjunct therapy in obesity management, pending the demonstration of central bioavailability and safety in humans. Future research should explore optimal dosing regimens, blood–brain barrier penetration, and combination strategies with established weight-loss interventions to fully assess the translational potential. Bromelain could be a potential agent in the prevention and treatment of metabolic diseases by modulating glucose transport and signaling pathways. However, new formulations and delivery systems are needed to enhance its efficacy and ensure safe usage.

## Figures and Tables

**Figure 1 cimb-47-00607-f001:**
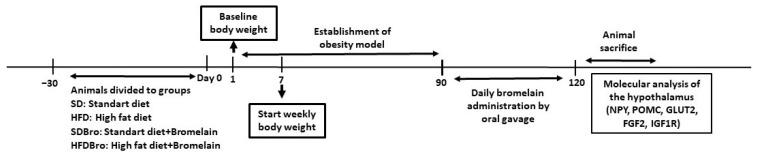
Schematic diagram of the experimental design: diet phases, bromelain administration, and sampling timeline. Abbreviations: NPY, neuropeptide Y; POMC, pro-opiomelanocortin; GLUT2, glucose transporter 2; FGF2, fibroblast growth factor 2; and IGF1R, insulin-like growth factor 1 receptor.

**Figure 2 cimb-47-00607-f002:**
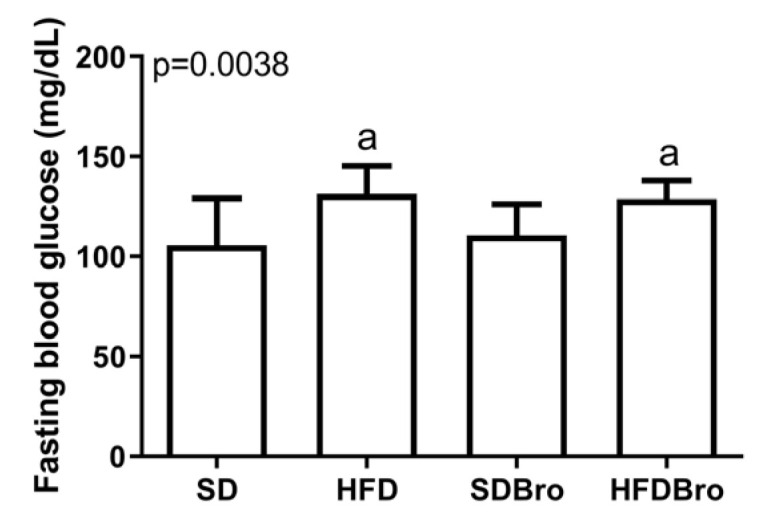
Fasting blood glucose levels at the end of treatment in standard diet (SD), standard diet + bromelain (SDBro), high-fat diet (HFD), and high-fat diet + bromelain (HFDBro) groups. Bars labeled with “a” differ significantly from the SD group (*p* < 0.05).

**Figure 3 cimb-47-00607-f003:**
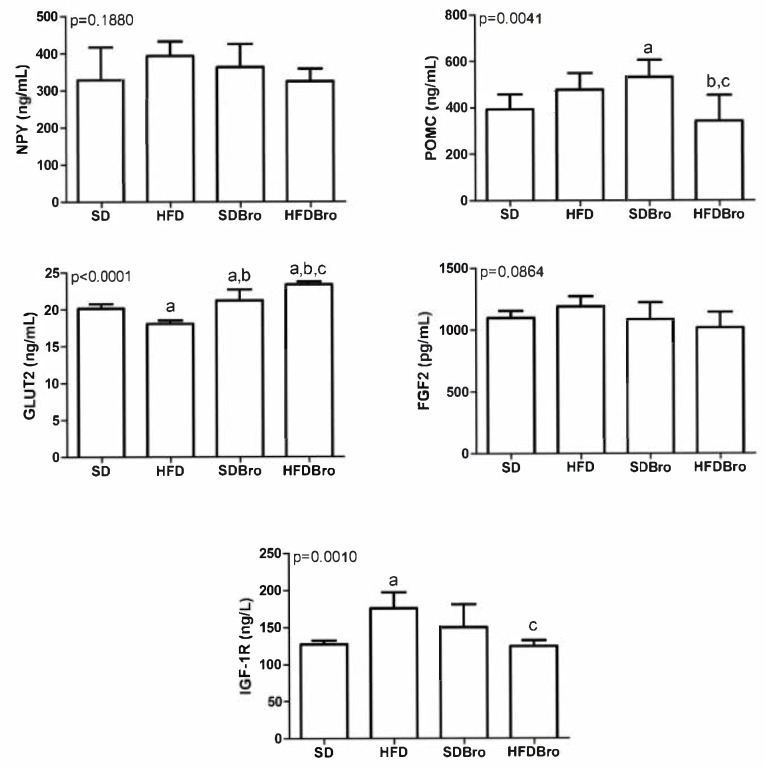
Hypothalamic IGF1R, GLUT2, POMC, NPY, and FGF2 expression at the end of treatment in standard diet (SD), standard diet + bromelain (SDBro), high-fat diet (HFD), and high-fat diet + bromelain (HFDBro) groups. Bars labeled with “a” differ significantly from the SD group, bars labeled with “b” differ significantly from the HFD group, and bars labeled with “c” differ significantly from the SDBro group (*p* < 0.05).

**Table 1 cimb-47-00607-t001:** Body weight changes over 90 days of dietary intervention.

Time (Days)	SD	HFD	SDBro	HFDBro	*p*-Value
0	154.30 ± 34.95	162.40 ± 46.18	153.87 ± 23.35	165.38 ± 34.88	0.8828
10	194.40 ± 39.55 *	212.40 ± 42.47 *	197.50 ± 26.27 *	230.25 ± 37.71 ^a^ *	0.2021
20	222.40 ± 48.04 *	273.50 ± 49.64 *	226.00 ± 27.35 *	281.63 ± 39.67 ^a^ *	0.0084
30	239.90 ± 52.62	294.30 ± 50.68	249.13 ± 31.92 *	303.13 ± 39.68 ^a^	0.0108
40	258.30 ± 54.77	318.10 ± 48.80 ^a^ *	273.62 ± 36.35 *	318.75 ± 36.64 ^a^	0.0126
50	270.50 ± 57.23	335.20 ± 47.56 ^a^	282.63 ± 37.58	339.13 ± 39.64 ^a^	0.0051
60	276.10 ± 55.20	357.70 ± 47.48 ^a^	293.75 ± 36.77 ^b^	360.38 ± 39.88 ^a,b,c^	0.0004
70	281.80 ± 57.08	378.20 ± 47.46 ^a^	300.88 ± 33.18 ^b^	377.88 ± 42.26 ^a,c^	<0.0001
80	303.30 ± 61.30 *	383.80 ± 47.33 ^a^	303.75 ± 34.82 ^b^	377.50 ± 44.21 ^a,b,c^	0.0004
90	310.50 ± 65.87	413.20 ± 59.46 ^a^ *	319.38 ± 29.82 ^b^ *	386.88 ± 31.59 ^a^	0.0002

Standard diet (SD), standard diet + bromelain (SDBro), high-fat diet (HFD), and high-fat diet + bromelain (HFDBro) groups. Values marked with “a” differ significantly from the SD group, values marked with “b” differ significantly from the HFD group, and values marked with “c” differ significantly from the SDBro group (*p* < 0.05). * indicate a significance level of *p* < 0.05 compared to the previous measurement within the same group. Values were given as mean ± standard deviation. 0 = baseline; 90 = end of diet period.

## Data Availability

Data available upon request from the authors.

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
