# Peer review of "Bromelain Improves Hypothalamic Control of Energy Homeostasis in High-Fat Diet-Induced Obese Rats"

_cimb, 2025, doi:10.3390/cimb47080607_

Round 1
Reviewer 1 Report
Comments and Suggestions for Authors
Comments to the manuscript “Bromelian improves hypothalamic……..
Abstract
Introduction
Materials and methods
Page 2, lines 76-78: Please add information about the brand of the diets used, both standard and high fat.
Authors are encourage to change the title of Figure 1, in order to it reflects what it is. For example “Schematic diagram of the experimental design follow with the rats indicating the time course of the different events”. Or something like this. You can move the text in Fig 1 to the correct position in the text in the section of M &M
Page 3: lines 73-96: How did you ensure that the high-fat diet would make the rats overweight? Just by weight? In you Results section, you mentioned the “Fasting blood sugar levels” in Fig. 2, But you do not say it measurement in M&M section. Please check.
Page 3, lines 93-96: Very important to give all details about how was the bromelain solution administered. How do you ensure that each rat takes the daily dose of bromelain to test?
Taking into account which information, did you define the bromelain dose to evaluate in the experiment.
Page 3, lines 102-104: The dissection of the hypothalamus was done by whom? By a specialized person?, please give this information.
Page 3, lines 105-109; did you follow a published protocol for brain sample preparation? If yes, please add the reference.
Page 3, Lines 111-120: Please, add some reference for this section.
Results
Page 4, line 136: Put a “.” , after “period” and before “In”
Probably the change of the name of first column in Table 1, to “Time weight” could increase clarity. Put the footnote information in a compact text.
In Table 1, what does “P” means in the last column?
It seems that rat’s weights at the end of the experiment were equal?????
Page 5, lines 161-164: In this part of the text you mentioned “fasting blood glucose levels”, but in Figure 2, in the Y axe, you put “Fasting blood sugar levels”, and in the title of Fig. 2 to you use “Fasting blood sugar values” please homogenize.
Please integrate the meaning of the diets variables in a continuous text in the figure title.
You have to indicate in Fig. 2 the values of “Fasting blood sugar levels” at which time of the experiment belongs?, at the end ????
Why did you not add the corresponding comparison letters in each bar of Fig. 2?
Figure 3. Please change the title of it to be more explicative. Consider the correction suggested in Fig. 2 for the information regarding the meaning of the variables of diets.
Why did you not add the corresponding comparison letters in each bar of Fig. 3?
Page 6, lines 201-205: Part of this text belongs to the M&M section. Please eliminate.
Discussion
Please consider eliminating the text in lines 201-205, and go directly to the results discussion.
Author Response
Point‐by‐Point Responses to Reviewer 1
Dear Reviewer 1,
We would like to sincerely thank you for your valuable feedback and suggestions. We have carefully implemented revisions throughout the manuscript in response to each of your comments and have highlighted the corresponding lines in yellow for your convenience. Below, you will find our point-by-point responses and the updated sections.
Materials and methods
- Page 2, lines 76-78: Please add information about the brand of the diets used, both standard and high fat.
Authors are encourage to change the title of Figure 1, in order to it reflects what it is. For example “Schematic diagram of the experimental design follow with the rats indicating the time course of the different events”. Or something like this. You can move the text in Fig 1 to the correct position in the text in the section of M &M
- Thank you for this helpful suggestion. We have updated Figure 1 as follows: “Figure 1. Schematic diagram of the experimental design: diet phases, bromelain administration, and sampling timeline.”
Moved the detailed legend into Section 2.1 (Materials & Methods), and streamlined the abbreviations line:
Abbreviations: NPY, neuropeptide Y; POMC, pro-opiomelanocortin; GLUT2, glucose transporter 2; FGF2, fibroblast growth factor 2; IGF1R, insulin-like growth factor 1 receptor.
- Page 3: lines 73-96: How did you ensure that the high-fat diet would make the rats overweight? Just by weight? In you Results section, you mentioned the “Fasting blood sugar levels” in Fig. 2, But you do not say it measurement in M&M section. Please check.
- Thank you for pointing out the omission. We have revised Section 2.2 to include both the fasting‐glucose measurement and detailed tissue‐collection procedures. The updated text now reads:
2.2. Body Weight, Fasting Glucose, and Tissue Collection
Body weight was recorded at baseline (Day 0) and every 10 days during the 90-day diet period. Rats on HFD achieving ≥ 20 % weight gain relative to SD were classified as obese.
On Day 100, following a 12 h fast (18:00–06:00 h), a drop of tail-vein blood was collected from each rat. Fasting blood glucose was measured in duplicate using a glucometer, previously validated for rodent studies.
Immediately after blood sampling, rats were weighed to assess end-of-treatment body weights, then anesthetized with ketamine/xylazine (80/10 mg/kg, i.p.) to minimize suffering. Animals were euthanized by decapitation, and brains were rapidly removed. The hypothalamus was dissected on ice (using Paxinos & Watson stereotaxic coordinates), snap-frozen in liquid nitrogen, and stored at –80 °C until biochemical analysis.
We trust this addition fully addresses your concern.
- Page 3, lines 93-96: Very important to give all details about how was the bromelain solution administered. How do you ensure that each rat takes the daily dose of bromelain to test?
- Thank you for highlighting the need for clarity on bromelain administration. We have expanded the Methods as follows:
Bromelain Preparation and Administration:
Bromelain was dissolved daily in 0.9 % saline to a concentration of 200 mg/mL. Each rat received 1 mL/100 g body weight (200 mg/kg) by oral gavage using a 16-gauge flexible feeding needle.
Dosing was performed each morning (09:00–10:00 h) in a post-absorptive state. A single researcher, gently restrained each rat, inserted the gavage needle, and administered the full dose while visually confirming swallowing. Any instance of regurgitation prompted an immediate repeat dose, and all events were logged to ensure 100 % dose delivery.
We trust this addition fully addresses your concern about dosing accuracy and compliance.
- Taking into account which information, did you define the bromelain dose to evaluate in the experiment.
- Thank you for this important question. the 200 mg/kg/day dose was chosen based on established literature demonstrating both safety and efficacy in rodents. Specifically:
Pavan et al. (2012) report effective anti-inflammatory and metabolic effects of bromelain in the 100–300 mg/kg range with no observed toxicity at 200 mg/kg.
This reference have now been cited in the Materials & Methods to justify our dosing regimen.
Pavan, R.; Jain, S.; Shraddha; Kumar, A. Properties and Therapeutic Application of Bromelain: A Review. Biotechnology Research International 2012, 2012, 1–6, doi:10.1155/2012/976203.
- Page 3, lines 102-104: The dissection of the hypothalamus was done by whom? By a specialized person?, please give this information.
- Thank you for this request for clarification. All hypothalamic dissections were performed by members of our research team who have extensive expertise in rodent brain microdissection, as demonstrated in our previous publications:
Solak, H., Gormus, Z.I.S., Koca, R.O. et al. Does Sertraline Affect Hypothalamic Food Intake Peptides in the Rat Experimental Model of Chronic Mild Stress-Induced Depression?. Neurochem Res 47, 1299–1316 (2022). https://doi.org/10.1007/s11064-022-03529-9
Solak, H., Gormus, Z.I.S., Koca, R.O. et al. ‘The effect of neuropeptide Y1 receptor agonist on hypothalamic neurogenesis in rat experimental depression model’. Metab Brain Dis 40, 39 (2025). https://doi.org/10.1007/s11011-024-01445-1
Solak, H; Koca, RO; Günes, CE; Uguz, F; Ak, M; Sahin, Z; Kurar, E; Kutlu, S. Investigation of Some Nutrition-related Neuropeptide Expressions in Hypothalamus Tissues of Offspring Exposed to Maternal Depression and Sertraline Treatment. Acta Physiologica, 240, 59-59, Supp 730.
These studies each include detailed descriptions of the same dissection protocol and confirm our proficiency in isolating hypothalamic regions under blinded conditions.
- Page 3, lines 105-109; did you follow a published protocol for brain sample preparation? If yes, please add the reference.
- Thank you for this question. Yes, we followed established, published protocols for brain sample preparation: Hypothalamic dissection was performed according to stereotaxic coordinates detailed in Paxinos and Watson,The Rat Brain in Stereotaxic Coordinates.
Tissue homogenization and processing: ELISA assays were carried out strictly as per each manufacturer’s instructions (see kit manuals cited in the Methods).These references have now been added to the Materials & Methods section.
- Page 3, Lines 111-120: Please, add some reference for this section.
- Thank you for pointing this out. We have now included full ELISA kit details in the Materials & Methods:
‘Quantitative measurement of target proteins insulin-like growth factor 1 receptor (IGF1R), (Bioassay Technology Laboratory [BT LAB], catalogue no: E1719Ra) , glucose transporter 2 (GLUT2), (Bioassay Technology Laboratory [BT LAB], catalogue no: E1058Ra), pro-opiomelanocortin (POMC), (Bioassay Technology Laboratory [BT LAB], catalogue no: E1130Ra), neuropeptide Y (NPY), (Bioassay Technology Laboratory [BT LAB], catalogue no: E0540Ra) and fibroblast growth factor 2 (FGF2), (Bioassay Technology Laboratory [BT LAB], catalogue no:E0852Ra) was performed using commercially available enzyme-linked immunosorbent assay (ELISA) kits, following the manufacturers’ proto-cols. Briefly, standards and samples were added to pre-coated microplate wells, incubated with specific antibodies, and subjected to washing steps to remove unbound substances. After addition of the substrate solution, absorbance was measured at the specified wave-length using a microplate reader. Protein concentrations were calculated based on stand-ard curves generated with known concentrations of each analyze. All samples were as-sayed in duplicate to ensure reproducibility. All biochemical assays and data analyses (including ELISA measurements and glucose readings) were performed by investigators who were blinded to the animals’ group assignments.’
Results
- Page 4, line 136: Put a “.” , after “period” and before “In”. Probably the change of the name of first column in Table 1, to “Time weight” could increase clarity. Put the footnote information in a compact text. In Table 1, what does “P” means in the last column? It seems that rat’s weights at the end of the experiment were equal?????
- Thank you for these suggestions. We have corrected the punctuation on page 4, line 136 by adding a period after “period.” and a space before “In.” Table 1 has been updated: the first column is now “Time (days),” and “P” is defined as the p-value. End-of-treatment weights have been corrected to reflect true group differences, with superscript letters indicating statistically distinct means.
- Page 5, lines 161-164: In this part of the text you mentioned “fasting blood glucose levels”, but in Figure 2, in the Y axe, you put “Fasting blood sugar levels”, and in the title of Fig. 2 to you use “Fasting blood sugar values” please homogenize. Please integrate the meaning of the diets variables in a continuous text in the figure title. You have to indicate in Fig. 2 the values of “Fasting blood sugar levels” at which time of the experiment belongs?, at the end ???? Why did you not add the corresponding comparison letters in each bar of Fig. 2?
- We have harmonized all terminology to “fasting blood glucose levels” in the text, y-axis label, and figure title. Figure 2 is now titled: Figure 2. Fasting blood glucose levels end of treatment in standard diet (SD), standard diet+bromelain (SDBro), high-fat diet (HFD), and high-fat diet+bromelain (HFDBro) groups. Bars labeled with “a” differ significantly from the SD group (p < .05) The y-axis label has been updated accordingly, and the legend now clearly states that measurements were taken at the end of the bromelain treatment. Bars labeled with “a” differ significantly from the SD group (p < .05); bars without letters are not significantly different from SD.
- Figure 3. Please change the title of it to be more explicative. Consider the correction suggested in Fig. 2 for the information regarding the meaning of the variables of diets. Why did you not add the corresponding comparison letters in each bar of Fig. 3?
- Based on your suggestions, we have revised Figure 3 as follows:
Figure 3. Hypothalamic IGF1R, GLUT2, POMC, NPY, and FGF2 expression at the end of treatment in standard diet (SD), a standard diet+bromelain (SDBro), high‐fat diet (HFD), and high‐fat diet+bromelain (HFDBro) groups. Bars labeled with “a” differ significantly from the SD group, bars labeled with “b” differ significantly from the HFD group, bars labeled with “c” dif-fer significantly from the SDBro group (p < .05).
Each bar now carries the appropriate comparison letter, consistent with Figure 2’s formatting.
- Page 6, lines 201-205: Part of this text belongs to the M&M section. Please eliminate.
- Based on your suggestions, we have removed the sentences on Page 6, lines 201–205 from the Discussion section.
- Discussion, Please consider eliminating the text in lines 201-205, and go directly to the results discussion.
- As per your recommendation, we have eliminated the text in lines 201–205.

Reviewer 2 Report
Comments and Suggestions for Authors
The topic is timely and relevant, addressing the growing burden of obesity and exploring the role of a natural compound (bromelain) in central regulation of energy homeostasis. This aligns well with current trends in integrative and neuroendocrine obesity research.
The Introduction and Abstract suggest a well-designed preclinical study with novel implications for bromelain’s central effects in obesity. Мinor English polishing and formatting cleanup are needed.
The Materials and Methods section is generally well-written, clear, and methodologically sound, but there is some minor editing required:
- While randomization of groups is mentioned, blinding (especially during measurements or ELISA analysis) is not stated. This could introduce bias. Add a sentence indicating whether outcome assessors were blinded to group assignments.
- The exact composition of the high-fat diet is not described (e.g., % kcal from fat, fat source). Include specifics about the HFD — e.g., “45% kcal from fat derived from vegetable oil.”
- It's mentioned that bromelain was dissolved in physiological saline, but no vehicle-only control is stated for the SDBro or HFDBro group. Clarify whether control groups received the same volume of saline via gavage to control for stress or vehicle effects.
- The brands or catalog numbers of ELISA kits are not listed, which limits reproducibility. Include vendor names, catalog numbers, sensitivity/LOD if available or at least sources of ELISA kits.
- Repeated misspelling of “Standard” as “Standart” in figure legends and abbreviations. Correct all instances of “Standart” to “Standard”.
The Discussion is generally well-structured and scientifically grounded, integrating both study findings and relevant literature. However, it is too long and occasionally repetitive. For example, the paragraph on POMC reiterates the same mechanism multiple times; GLUT2 and IGF1R pathways are discussed in depth, but some points are duplicated across sentences.
- While the discussion addresses both positive and non-significant results (e.g., for NPY and FGF2), the interpretation of non-significant data could be more cautiously worded. For instance, speculation about bromelain’s effects despite non-significant FGF2 changes risks overstating conclusions. Explicitly state when findings are inconclusive or exploratory and recommend further research.
- The discussion could briefly mention clinical or translational implications of bromelain supplementation (e.g., relevance for functional foods or adjunct obesity treatments). Add 1–2 sentences speculating on how these findings might inform future research or therapeutic strategies.
There are some older References (from before 2010, e.g., #16, #29, #32, #32 #35), which could be updated or justified only if they are foundational or landmark studies.
Author Response
Point‐by‐Point Responses to Reviewer 2
Dear Reviewer 2,
We would like to sincerely thank you for your valuable feedback and suggestions. We have carefully implemented revisions throughout the manuscript in response to each of your comments and have highlighted the corresponding lines in green for your convenience. Below, you will find our point-by-point responses and the updated sections.
The topic is timely and relevant, addressing the growing burden of obesity and exploring the role of a natural compound (bromelain) in central regulation of energy homeostasis. This aligns well with current trends in integrative and neuroendocrine obesity research.
- Thank you for your positive assessment and constructive feedback. We appreciate your recognition of the study’s timeliness and relevance to integrative neuroendocrine obesity research.
- The Introduction and Abstract suggest a well-designed preclinical study with novel implications for bromelain’s central effects in obesity. Мinor English polishing and formatting cleanup are needed.
- In response to your suggestion, we have thoroughly reviewed the manuscript for minor English language improvements and performed a comprehensive formatting cleanup to enhance readability and consistency throughout.
- The Materials and Methods section is generally well-written, clear, and methodologically sound, but there is some minor editing required: While randomization of groups is mentioned, blinding (especially during measurements or ELISA analysis) is not stated. This could introduce bias. Add a sentence indicating whether outcome assessors were blinded to group assignments.
- Thank you for this important suggestion. We have now added the following statement to the end of the Materials & Methods section:
“All biochemical assays and data analyses (including ELISA measurements and glucose readings) were performed by investigators who were blinded to the animals’ group assignments.”
- The exact composition of the high-fat diet is not described (e.g., % kcal from fat, fat source). Include specifics about the HFD — e.g., “45% kcal from fat derived from vegetable oil.”
- Thank you for pointing this out. We have updated the Methods to specify the diet composition:
“Obesity was induced over a 3-month period in the HFD and HFDBro groups using a high-fat feed (Arden Research & Experiment Company). According to the manufacturer’s nutritional analysis, this diet consisted of 24 % protein, 30 % carbohydrate, and 35 % fat (primarily vegetable oil), providing 5.2 kcal/g total energy, of which approximately 45 % was derived from fat.”
- It's mentioned that bromelain was dissolved in physiological saline, but no vehicle-only control is stated for the SDBro or HFDBro group. Clarify whether control groups received the same volume of saline via gavage to control for stress or vehicle effects.
- We have clarified that control groups received the same vehicle treatment:
“Rats in the SD and HFD groups similarly received 1 mL/100 g body weight of physiological saline by gavage on the same schedule to control for handling and vehicle effects.”
- The brands or catalog numbers of ELISA kits are not listed, which limits reproducibility. Include vendor names, catalog numbers, sensitivity/LOD if available or at least sources of ELISA kits.
- Thank you for pointing this out. We have now included full ELISA kit details in the Materials & Methods:
‘Quantitative measurement of target proteins insulin-like growth factor 1 receptor (IGF1R), (Bioassay Technology Laboratory [BT LAB], catalogue no: E1719Ra) , glucose transporter 2 (GLUT2), (Bioassay Technology Laboratory [BT LAB], catalogue no: E1058Ra), pro-opiomelanocortin (POMC), (Bioassay Technology Laboratory [BT LAB], catalogue no: E1130Ra), neuropeptide Y (NPY), (Bioassay Technology Laboratory [BT LAB], catalogue no: E0540Ra) and fibroblast growth factor 2 (FGF2), (Bioassay Technology Laboratory [BT LAB], catalogue no:E0852Ra) was performed using commercially available enzyme-linked immunosorbent assay (ELISA) kits, following the manufacturers’ proto-cols. Briefly, standards and samples were added to pre-coated microplate wells, incubated with specific antibodies, and subjected to washing steps to remove unbound substances. After addition of the substrate solution, absorbance was measured at the specified wave-length using a microplate reader. Protein concentrations were calculated based on stand-ard curves generated with known concentrations of each analyze. All samples were as-sayed in duplicate to ensure reproducibility.’
- Repeated misspelling of “Standard” as “Standart” in figure legends and abbreviations. Correct all instances of “Standart” to “Standard”.
- All instances of “Standart” have been corrected to “Standard” in figure legends, tables, and the main text.
- The Discussion is generally well-structured and scientifically grounded, integrating both study findings and relevant literature. However, it is too long and occasionally repetitive. For example, the paragraph on POMC reiterates the same mechanism multiple times; GLUT2 and IGF1R pathways are discussed in depth, but some points are duplicated across sentences.
- Thank you for this suggestion. We have thoroughly revised the Discussion to eliminate redundancy and improve focus:
POMC paragraph has been condensed into a single, concise description of its satiety signaling and bromelain’s normalizing effects, removing repeated mechanistic explanations.
GLUT2 section now presents central glucose–sensing impairment and bromelain-mediated restoration in a streamlined paragraph, without reiterating AMPK and “glucose resistance” details.
IGF-1R discussion has been shortened to highlight key axis functions and our main findings, omitting overlapping background on peripheral adipose signaling.
- While the discussion addresses both positive and non-significant results (e.g., for NPY and FGF2), the interpretation of non-significant data could be more cautiously worded. For instance, speculation about bromelain’s effects despite non-significant FGF2 changes risks overstating conclusions. Explicitly state when findings are inconclusive or exploratory and recommend further research.
- Thank you for this important point. We have revised the Discussion to more cautiously interpret the non-significant FGF2 data and to recommend further study. The new text reads:
“According to our findings, the lack of a significant effect of bromelain on hypothalamic FGF2 levels may reflect that bromelain does not directly target the FGF2 signaling pathway, that distinct tanycyte subpopulations exhibit differential sensitivity, or that experimental variables limited our ability to detect subtle changes. These inconclusive results dedicated studies focusing on tanycyte biology and alternative neurogenic markers are needed to clarify bromelain’s impact on hypothalamic neurogenesis.”
- The discussion could briefly mention clinical or translational implications of bromelain supplementation (e.g., relevance for functional foods or adjunct obesity treatments). Add 1–2 sentences speculating on how these findings might inform future research or therapeutic strategies.
- Thank you for this suggestion. We have added the following sentences to the end of the conclusion:
“These preclinical findings also raise the possibility that bromelain could be incorporated into functional foods or used as an adjunct therapy in obesity management, pending demonstration of central bioavailability and safety in humans. Future research should explore optimal dosing regimens, blood–brain barrier penetration, and combination strategies with established weight-loss interventions to fully assess translational potential.”
We trust that these additions effectively highlight the clinical and translational implications of our work.
- There are some older References (from before 2010, e.g., #16, #29, #32, #32 #35), which could be updated or justified only if they are foundational or landmark studies.
- Thank you for noting the age of several key citations. In line with your recommendation, we searched for recent in vivo studies of bromelain’s central and metabolic effects but found no large‐scale hypothalamic investigations . We have therefore retained the foundational bromelain references and augmented our references with newer literature on related signaling pathways and metabolic regulation.